# Positive selection and relaxed purifying selection contribute to rapid evolution of male-biased genes in a dioecious flowering plant

Lei Zhao[1†], Wei Zhou[1†], Jun He[1], De-Zhu Li[1,2]*, Hong-Tao Li[1,2]*

[1]Germplasm Bank of Wild Species & Yunnan Key Laboratory of Crop Wild Relatives Omics, Kunming Institute of Botany, Chinese Academy of Sciences, Kunming, Yunnan, China; [2]Kunming College of Life Science, University of Chinese Academy of Sciences, Kunming, China

*For correspondence:
dzl@mail.kib.ac.cn (DZL);
lihongtao@mail.kib.ac.cn (HTL)

[†]These authors contributed equally to this work

**Abstract** Sex-biased genes offer insights into the evolution of sexual dimorphism. Sex-biased genes, especially those with male bias, show elevated evolutionary rates of protein sequences driven by positive selection and relaxed purifying selection in animals. Although rapid sequence evolution of sex-biased genes and evolutionary forces have been investigated in animals and brown algae, less is known about evolutionary forces in dioecious angiosperms. In this study, we separately compared the expression of sex-biased genes between female and male floral buds and between female and male flowers at anthesis in dioecious *Trichosanthes pilosa* (Cucurbitaceae). In floral buds, sex-biased gene expression was pervasive, and had significantly different roles in sexual dimorphism such as physiology. We observed higher rates of sequence evolution for male-biased genes in floral buds compared to female-biased and unbiased genes. Male-biased genes under positive selection were mainly associated with functions to abiotic stress and immune responses, suggesting that high evolutionary rates are driven by adaptive evolution. Additionally, relaxed purifying selection may contribute to accelerated evolution in male-biased genes generated by gene duplication. Our findings, for the first time in angiosperms, suggest evident rapid evolution of male-biased genes, advance our understanding of the patterns and forces driving the evolution of sexual dimorphism in dioecious plants.

## eLife assessment

This **valuable** paper examines gene expression differences between male and female individuals over the course of flower development in the dioecious angiosperm Trichosantes pilosa. Male-biased genes evolve faster than female-biased and unbiased genes, which is frequently observed in animals, but this is the first report of such a pattern in plants. In spite of the limited sample size, the evidence is mostly **solid** and the methods appropriate for a non-model organism. The resources produced will be used by researchers working in the Cucurbitaceae, and the results obtained advance our understanding of the mechanisms of plant sexual reproduction and its evolutionary implications: as such they will broadly appeal to evolutionary biologists and plant biologists.

## Introduction

Sexual dimorphism is the condition where sexes of the same species exhibit different morphological, ecological, and physiological traits in gonochoristic animals and dioecious plants, despite male

and female individuals sharing the same genome except for sex chromosomes or sex-determining loci (*Mank, 2009*; *Barrett and Hough, 2013*). Such sexual dimorphisms usually arise from differential expression of genes between the two sexes, i.e., sex-biased genes (including sex-specific genes expressed exclusively in one sex) that are located on autosomal chromosomes and sex chromosomes/or sex-determining regions (*Ellegren and Parsch, 2007*; *Parsch and Ellegren, 2013*; *Grath and Parsch, 2016*; *Charlesworth, 2018*; *Tosto et al., 2023*). Recently, some studies have begun to explore the strength and impact of evolutionary forces that shape different sexually dimorphic traits through sex-biased gene expression (*Mank, 2009*; *Rowe et al., 2018*; *Naqvi et al., 2019*; *Cai et al., 2021*; *Mank, 2023*; *Murat et al., 2023*; *Singh et al., 2023*). Previous studies revealed that sex-biased gene expressions were associated with the evolution of sexual dimorphisms in some animal species, although the extent of this bias exhibits great variation among taxa, tissues, and development stages (*Mank, 2017*; *Harrison et al., 2015*; *Hsu et al., 2020*; *Khodursky et al., 2020*; *Lichilín et al., 2021*; *Toubiana et al., 2021*; *Djordjevic et al., 2022*; *Yue et al., 2023*). Unlike most animals, the vast majority (~90%) of flowering plants (angiosperms) are hermaphroditic, while only a small fraction (~5%) are dioecious in which individuals have exclusively male or female reproductive organs (*Renner, 2014*). Most dioecious plants possess homomorphic sex-chromosomes that are roughly similar in size when viewed by light microscopy (*Palmer et al., 2019*). Furthermore, sexual dimorphism in dioecious plants is less common and less conspicuous than in most animals (*Barrett and Hough, 2013*). Hence, the study of sex-biased gene expression is of great interest to plant evolutionary biologists, as it is necessary to understand the evolution of sexual dimorphism in dioecious plants (*Moore and Pannell, 2011*).

A common pattern that has emerged from previous studies is that sex-biased genes, particularly male-biased genes, tend to evolve rapidly in protein sequence (the ratio of non-synonymous to synonymous substitutions, $d_N/d_S$) compared to unbiased genes (*Ellegren and Parsch, 2007*; *Grath and Parsch, 2016*). The rapid evolution of male-biased genes was first observed in *Drosophila melanogaster* (*Zhang et al., 2004*; *Zhang and Parsch, 2005*) and has been supported by recent investigations in a wider range of animals (*Pröschel et al., 2006*; *Mank et al., 2007*; *Mank, 2017*; *Papa et al., 2017*; *Catalán et al., 2018*; *Toubiana et al., 2021*). In recent years, there have been growing studies on the expression dynamics and molecular evolutionary rates of sex-biased genes in flowering plants, including hermaphroditic *Arabidopsis thaliana* (*Gossmann et al., 2014*; *Gossmann et al., 2016*), *Solanum* (*Moyle et al., 2021*), and dioecious *Silene latifolia* (*Zemp et al., 2016*), *Salix viminalis* (*Darolti et al., 2018*), *Mercurialis annua* (*Cossard et al., 2019*), *Populus balsamifera* (*Sanderson et al., 2019*), and *Leucadendron* (*Scharmann et al., 2021*). However, despite such advances, the molecular evolution pattern of sex-biased genes in plants remains inconsistent among the studied plant species (*Muyle, 2019*; *Veltsos, 2019*). In dioecious plants such as *Mercurialis annua* and *Leucadendron*, *Cossard et al., 2019* and *Scharmann et al., 2021* found no significant differences in evolutionary rates of proteins among female-biased, male-biased, and unbiased genes detected between male and female plants leaf tissues, although the expression of sex-biased genes was highly different from unbiased genes in leaves. Similar patterns have also been reported in dioecious *Populus balsamifera*, where evolutionary rates of male-biased, female-biased, and unbiased genes did not differ in reproductive tissues (*Sanderson et al., 2019*). However, in the dioecious *Salix viminalis*, male-biased genes have significantly lower evolutionary rates of proteins than female-biased and unbiased genes in catkin tissues (*Darolti et al., 2018*). To our knowledge, only the five above-mentioned studies have investigated expression differences and protein evolutionary rates of sex-biased genes in dioecious angiosperms. Moreover, these studies only compared gene expression in vegetative versus vegetative tissues and vegetative versus reproductive tissues, limiting our understanding of sexual selection at different floral development stages. Therefore, more studies and taxa are needed to explore the common patterns of sequence evolution in sex-biased genes, with more focus on comparing sex-biased gene expression in reproductive versus reproductive tissues, e.g., different floral development stages in dioecious angiosperms.

Evolutionary analyses indicate that different driving forces impact the rate of sequence evolution of sex-biased genes. These forces include positive selection, which promotes the spread and adaptive fixation of beneficial alleles; sexual selection, which results from male-male competition or female choice; and relaxed purifying selection, which reduces the removal of deleterious mutations (*Grath and Parsch, 2016*; *Mank, 2017*; *Dapper and Wade, 2020*). For example, in animal systems,

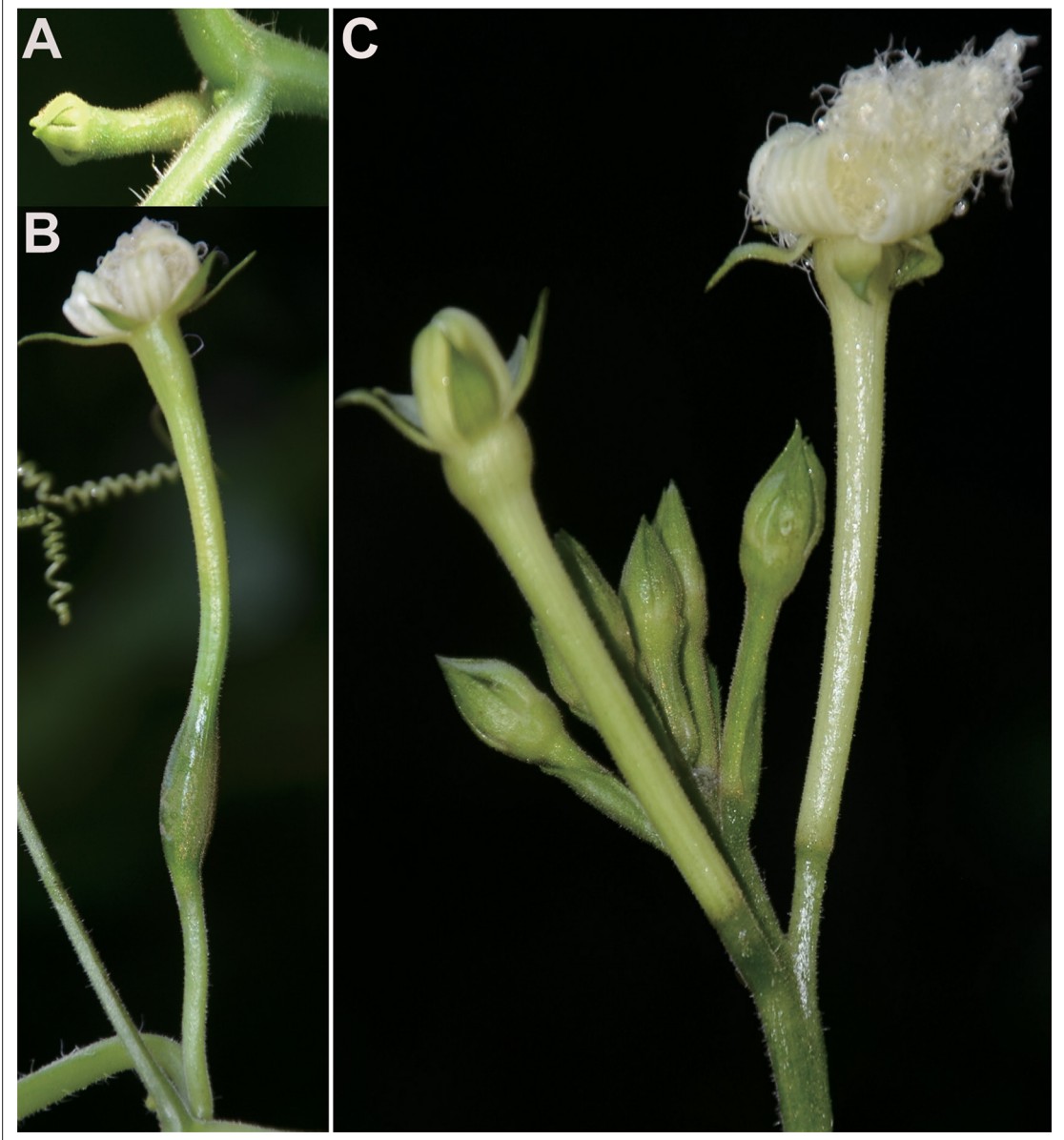

**Figure 1.** Floral buds and flowers at anthesis of females (**A, B**) and males (**C**) in *Trichosanthes pilosa*.

particularly in *Drosophila*, the elevated sequence divergence rates of male-biased genes have often been interpreted as the signature of adaptive evolution, suggesting that sexual selection is the primary evolutionary force (**Pröschel et al., 2006** ; **Assis et al., 2012**). In brown algae, female-biased and/or male-biased genes exhibited higher evolutionary rates than unbiased genes, suggesting that rapid evolution is partly driven by adaptive evolution or sexual selection (**Lipinska et al., 2015**; **Cossard et al., 2022**; **Hatchett et al., 2023**). However, studies in plants have never reported elevated rates of sex-biased genes.

An alternative explanation for the rapid evolution of sex-biased genes is a relaxation of purifying selection due to reduced constraints (**Lahti et al., 2009**; **Dapper and Wade, 2020**). In the model plant *Arabidopsis thaliana*, pollen genes were found to be evolving faster than sporophyte-specific genes due to relaxed purifying selection associated with the transition from outcrossing to selfing (**Harrison et al., 2019**). These trends were recently confirmed in *Arabis alpina*, which exhibits mating system variation across its distribution, suggesting that the efficacy of purifying selection on male gametophyte-expressed genes was significantly weaker in inbred populations (**Gutiérrez-Valencia**

*et al., 2022*). Together, these findings in plants reinforce the idea that both adaptive (e.g. positive selection, sexual selection) and non-adaptive (e.g. relaxed selection) evolutionary processes differentially impact the sequence evolution of sex-biased genes. Hence, investigating the potential contribution of selection forces to the emergence of specific evolutionary patterns of sex-biased genes within a focal species is of great interest.

In the family Cucurbitaceae, there are about 96 genera and 1000 species, about 50% of species are dioecious, and the others are monoecious (*Schaefer and Renner, 2011*). Phylogenetic analyses of Cucurbitaceae suggest that dioecy is the ancestral state of the family, but transitions frequently to monoecy (*Zhang et al., 2006*). *Trichosanthes pilosa* (synonym: *T. ovigera*, 2n=22, Cucurbitaceae) is mainly distributed from Southwest and Southeast China to Japan, extending to Southeast Asia, New Guinea and Western Australia. It was suggested to have originated in the late Miocene (ca. 8–6 million-year ago) (*de Boer et al., 2012*; *de Boer et al., 2015*; *Guo et al., 2020*). *Trichosanthes pilosa* is a perennial, night-flowering, insect-pollinated dioecious vine that reproduces sexually and possesses a pair of heteromorphic sex chromosomes XX/XY (*Ming et al., 2011*). The male parts (e.g. anthers) of female flowers, and the female parts (e.g. pistil and ovaries) of male flowers are fully aborted. Its male and female flowers exhibit strong sexual dimorphism in floral morphological and phenological traits, such as racemose versus solitary (*Figure 1*), early-flowering versus late-flowering, and caducous versus long-lived (*Wu et al., 2011*).

To understand the evolution of sex-biased genes in dioecious *T. pilosa*, we collected floral buds and flowers at anthesis from male and female individuals and characterized their expression profiles using Illumina RNA sequencing. Our primary objectives are to (1) compare expression divergence between males and females at two floral development stages; (2) explore whether there are differences in the evolutionary rates of proteins among female-biased, male-biased, and unbiased genes; and if so, (3) determine the main selective forces that contribute to the differentiation of sequence evolution rates among gene categories.

## Results

### Transcriptome sequencing, de novo assembly, and annotation

Using whole transcriptome shotgun sequencing, we sequenced floral buds and flowers at anthesis from females and males of dioecious *T. pilosa*. We set up three biological replicates from three female and three male plants, including 12 samples in total (six floral buds and six flowers at anthesis). We then generated a total of nearly 276 million clean reads (*Supplementary file 1*). Due to the absence of a reference genome, we performed de novo assembly of transcripts from all the clean reads, followed by clustering and filtering analysis, resulting in 59,051 unigenes (*Figure 2—figure supplement 1*). To evaluate the quality of the assembled unigenes, we used BUSCO assessments based on embryophyta_odb10 database, which showed the completeness of the reference transcriptome at 89.7% (*Supplementary file 2*). We then annotated them against protein databases including NR, KEGG, Swissport, PFAM, and GO using BLASTP and nucleotide database NT using BLASTN (*Supplementary file 2*). The e-value distribution of the best hits in the NR database suggested that 47,241 unigenes (80%) had strong homology, with an e-value smaller than 1.0e-15 (*Figure 2—figure supplement 1*). The majority of unigenes were annotated by homologs in species of Cucurbitaceae (61.6%, 36,375), such as *Momordica charantia* (16.3%, 9625), *Cucumis melo* (11.9%, 7027), *Cucurbita pepo* (11.9%, 7027), *Cucurbita moschata* (11.5%, 6791), *Cucurbita maxima* (10.1%, 5964), and other species (38.4%, 22,676) (*Figure 2—figure supplement 1*). Overall, our assessment suggested that we have generated high-quality reference transcriptomes.

### Expression characteristics of sex-biased genes

We mapped the RNA-seq reads of floral buds and flowers at anthesis onto the reference transcriptome in dioecious *T. pilosa*, which resulted in approximately 75% read mappings per sample (*Supplementary file 3*). In floral buds, we identified 5096 (9.50%) female-biased genes and 4214 (7.86%) male-biased genes (*Figure 2A*). In contrast, only 380 (0.70%) female-biased genes and 233 (0.43%) male-biased genes were detected in flowers at anthesis (*Figure 2B*). Using hierarchical clustering analysis, we evaluated different levels of gene expression across sexes and tissues (*Figure 2C*). Gene expression for female floral buds clustered most distantly from expression in female flowers

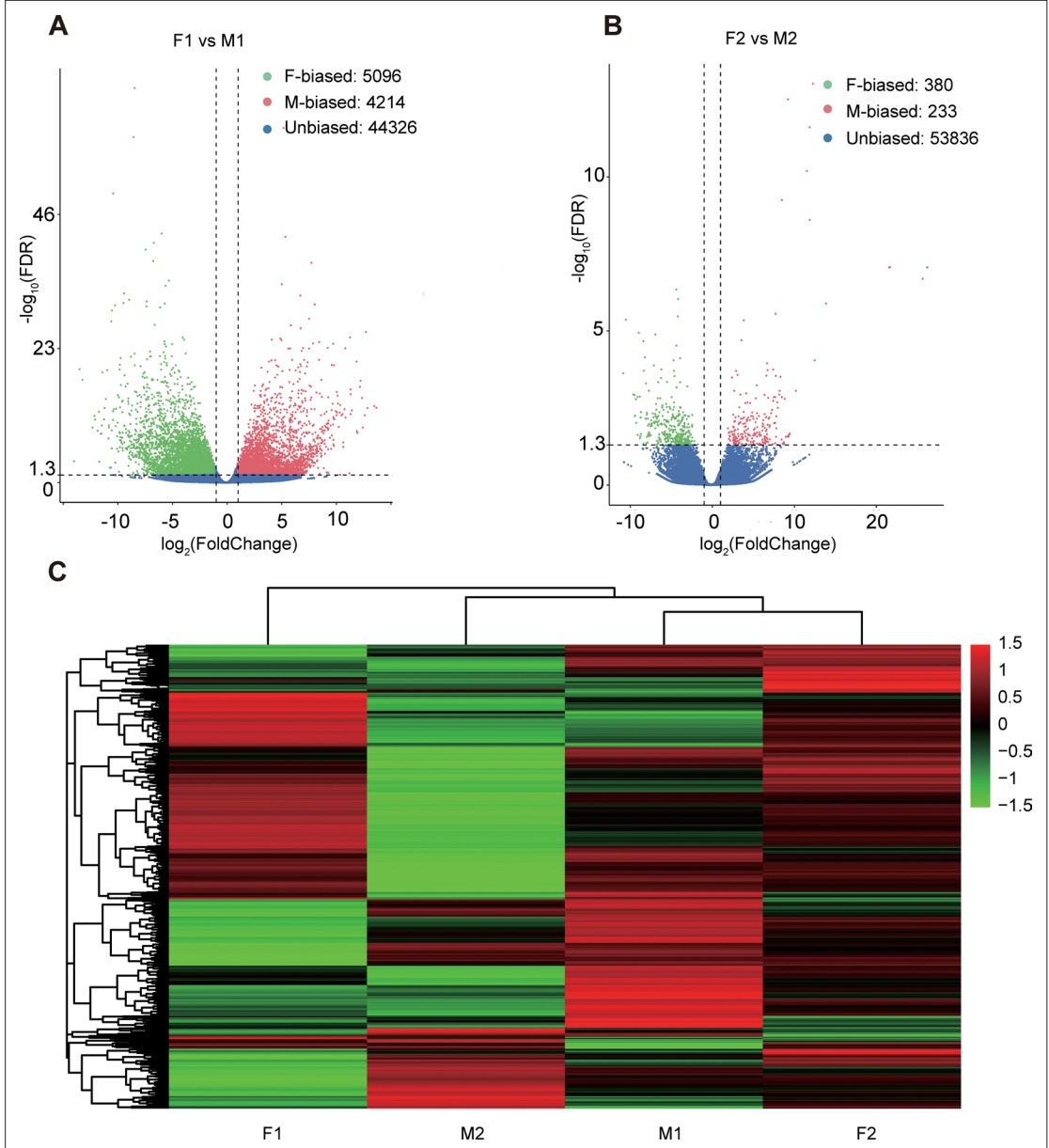

**Figure 2.** Sex-biased gene expression for floral buds and flowers at anthesis in males and females of *Trichosanthes pilosa*. Volcano plots of average expression between female-biased, male-biased and unbiased genes in floral buds (**A**) and flowers at anthesis (**B**). M1 and F1 indicate male and female floral buds; M2 and F2 indicate male and female flowers at anthesis. The value of y coordinate represents -$\log_{10}$(FDR), and the value of x coordinate represents $\log_2$(Fold Change) identified by DESeq2. Heatmap of sex-biased gene expression (**C**) using hierarchical clustering analysis. Hierarchical gene clustering is based on Euclidean distance with an average of $\log_2$(FPKM) for differentially expressed genes. The color gradient represents from high to low (from red to green) gene expression.

The online version of this article includes the following figure supplement(s) for figure 2:

**Figure supplement 1.** The length distribution of unigenes (**A**) and the e-value distribution (**B**) and the species distribution (**C**) of BLAST hits for each unigene.

at anthesis. However, expression in male floral buds clustered with expression in female flowers at anthesis, suggesting that male floral buds maybe tend to feminization in the early stages of floral development. Furthermore, we observed that the number of sex-biased genes in floral buds was approximately 15 times higher than in flowers at anthesis, indicating that sex-biased genes associated with meiotic processes, sex differentiation, and sexually dimorphic traits are predominantly expressed in floral buds. We also analyzed sex-specific genes that were exclusively expressed in floral buds and

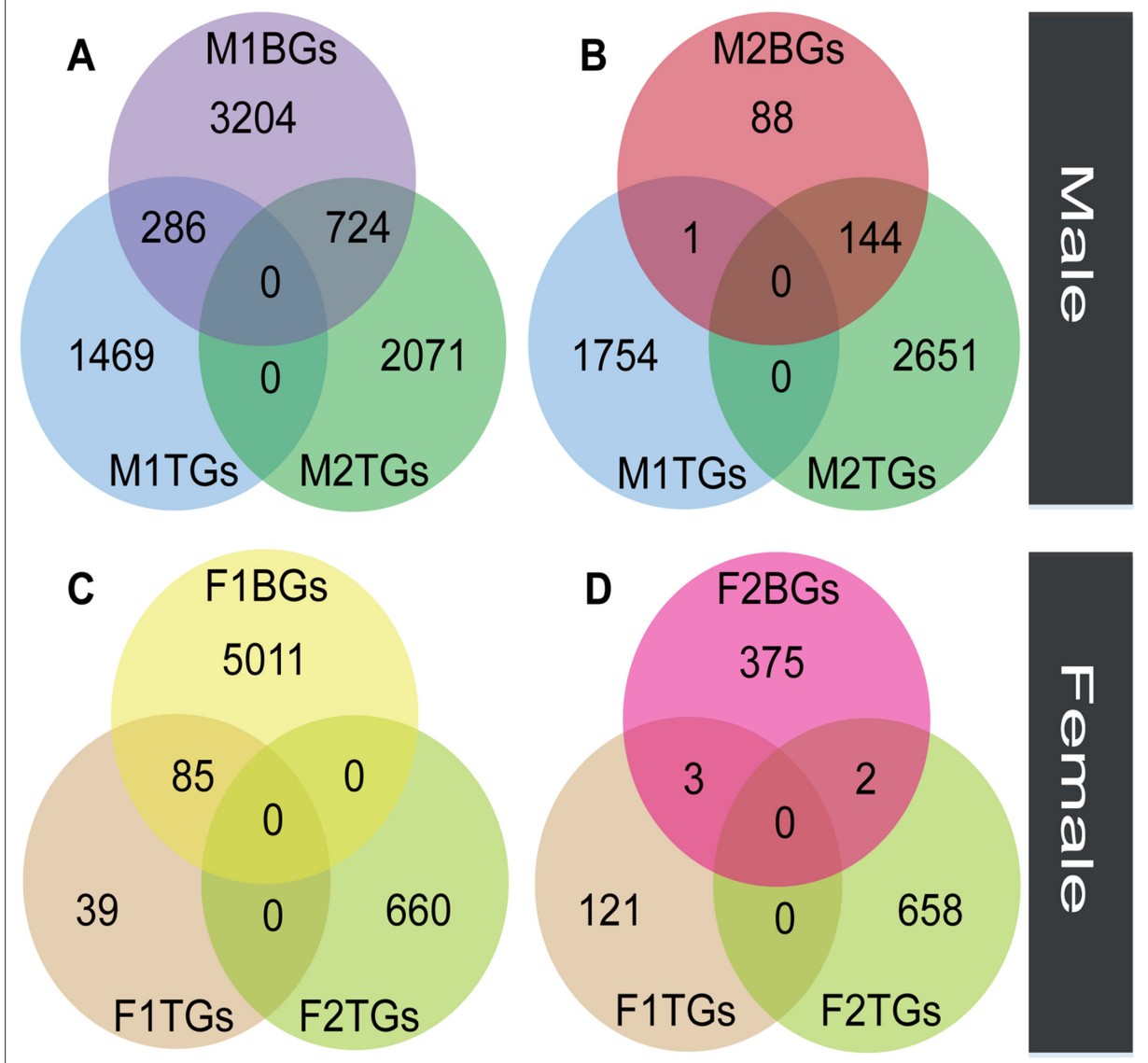

**Figure 3.** The overlap between sex-biased and tissue-biased genes in two types of sexes and tissues. Male-biased genes in floral buds (M1BGs) (**A**) or flowers at anthesis (M2BGs) (**B**) overlapped with tissue-biased genes in floral buds (M1TGs) and flowers at anthesis (M2TGs). Female-biased genes in floral buds (F1BGs) (**C**) or flowers at anthesis (F2BGs) (**D**) overlapped with tissue-biased genes in floral buds (F1TGs) and flowers at anthesis (F2TGs).

The online version of this article includes the following figure supplement(s) for figure 3:

**Figure supplement 1.** The overlap between male-biased genes with faster evolutionary rates and tissue-biased genes in floral buds.

flowers at anthesis of one sex. In floral buds, we found 253 out of 5096 (4.96%) female-specific genes and 465 out of 4214 (11.03%) male-specific genes. However, in flowers at anthesis, we only identified 26 out of 380 (6.84%) female-specific genes and 52 out of 233 (22.32%) male-specific genes. Taken together, sex bias is more prevalent in floral buds than in flowers at anthesis.

## Tissue-biased/stage-biased gene expression

We compared the expression levels of transcripts in floral buds and flowers at anthesis within each sex to identify genes with tissue-biased expression. In male plants, the number (M2TGs: n=2795) of tissue-biased genes in male flowers at anthesis (M2TGs) was 1040 higher than that in male floral buds (M1TGs: n=1755, *Figure 3A and B*). However, in female plants, the number (F2TGs: n=660) of tissue-biased genes in female flowers at anthesis (F2TGs) was only 536 more than that in female floral buds (F1TGs: n=124, *Figure 3C and D*). Our results indicated that males had a higher tissue-bias relative

to females. We also identified sex-biased genes that were expressed in both types of tissues by comparing tissue-biased genes with male-biased and female-biased genes, respectively. Few female-biased genes in floral buds (F1BGs: n=5096) overlapped with tissue-biased genes in female floral buds (F1TGs: n=124) and female flowers at anthesis (F2TGs: n=660), accounting for only 85 out of 5096 (1.67%) (*Figure 3C*). Similarly, few female-biased genes in flowers at anthesis (F2BGs: n=380) overlapped with tissue-biased genes in female floral buds (F1TGs: n=124) and female flowers at anthesis (F2TGs: n=660), occupying around 5 out of 380 (1.32%) (*Figure 3D*). However, a significant proportion of male-biased genes in floral buds (M1BGs: n=4214) overlapped with tissue-biased genes in male floral buds (M1TGs: n=1755) and male flowers at anthesis (M2TGs: n=2795), with 1010 out of 4214 (23.97%) (*Figure 3A*). A high proportion of male-biased genes in flowers at anthesis (M2BGs: n=233) overlapped with tissue-biased genes in male floral buds (M1TGs: n=1755) and male flowers at anthesis (M2TGs: n=2795), 145 out of 233 (62.23%) (*Figure 3B*).

## Elevated protein evolutionary rates of male-biased genes in floral buds

We compared rates of protein evolution among male-biased, female-biased and unbiased genes in four species with phylogenetic relationships ((((*T. anguina*, *T. pilosa*), *T. kirilowii*), *Luffa cylindrica*), including dioecious *T. pilosa*, dioecious *T. kirilowii*, monoecious *T. anguina* in *Trichosanthes*, together with monoecious *Luffa cylindrica*. To do this, we used the transcriptomes described above for *T. pilosa*. We also collected transcriptomes of *T. kirilowii*, as well as genomes of *T. anguina* and *Luffa cylindrica* (see Methods Section). We identified 1145 female-biased, 343 male-biased, and 2378 unbiased one-to-one orthologous groups (OGs) from floral buds. Additionally, we detected 45 female-biased, 13 male-biased, and 3782 unbiased one-to-one OGs from mature flowers in all four species. To quantify the rates of protein sequences, we separately calculated $\omega$ values for each sex-biased and unbiased orthologous gene using 'two-ratio' and 'free-ratio' branch models in juvenile and mature flowers (*Figure 4* and *Figure 4—figure supplement 1*).

The two-ratio branch model, where the foreground (dioecious branches) has a different $\omega$ value relative to the background (all other branches), is better supported than the fixed-ratio branch model, where all branches are constrained to have the same $\omega$ value. In the results of the 'two-ratio' branch model, the median of $\omega$ values in female-biased, male-biased, and unbiased genes were 0.227, 0.257, and 0.230 in floral buds, respectively (*Figure 4A* and *Supplementary file 4*). We observed that male-biased genes had a 13.22% and 11.74% higher median than female-biased and unbiased genes in floral buds, respectively. The difference in the distribution of $\omega$ values between female-biased versus male-biased genes (p=0.0021) and male-biased versus unbiased genes (p=0.0051) was statistically significant in Wilcoxon rank sum tests. However, we did not find a significant difference in $\omega$ values between female-biased and unbiased genes in floral buds (Wilcoxon rank sum test, p=0.4618). In flowers at anthesis, the median of $\omega$ values for female-biased, male-biased, and unbiased genes were 0.269, 0.177, and 0.231, respectively (*Figure 4B* and *Supplementary file 4*). However, there was no statistically significant difference in the distribution of $\omega$ values using Wilcoxon rank sum tests for female-biased versus male-biased genes (p=0.0556), female-biased versus unbiased genes (p=0.0796), and male-biased versus unbiased genes (p=0.3296) possibly because of limited statistical power due to the low number of sex-biased genes in flowers at anthesis.

In free-ratios model, $\omega$ values are free to vary in each branch compared to fixed-ratio branch model and the two-ratio branch model. The 'free-ratio' branch model yielded interesting results. In floral buds, the median $\omega$ values for female-biased, male-biased, and unbiased genes were 0.222, 0.265, and 0.226, respectively (*Figure 4C* and *Supplementary file 5*). Male-biased genes had a significantly higher median relative to female-biased genes (19.37% higher, Wilcoxon rank sum test, p=0.0009) and unbiased genes (17.26% higher, Wilcoxon rank sum test, p=0.0004) in floral buds. However, there was no significant difference in $\omega$ values between female-biased and unbiased genes (Wilcoxon rank sum test, p=0.9862). In flowers at anthesis, the median $\omega$ values for female-biased, male-biased, and unbiased genes were 0.300, 0.148, and 0.227, respectively (*Figure 4D* and *Supplementary file 5*). Female-biased and unbiased genes had significantly higher $\omega$ values than male-biased (Wilcoxon rank sum test, p=0.0101, p=0.0146, respectively). However, there was no significant difference in $\omega$ values between female-biased and unbiased genes (Wilcoxon rank sum test, p=0.2887). Since the number of male-biased genes and evolutionary rates of male-biased genes in flowers at anthesis are lower than those in floral buds, we decided to focus on the latter in subsequent analyses. Additionally, we

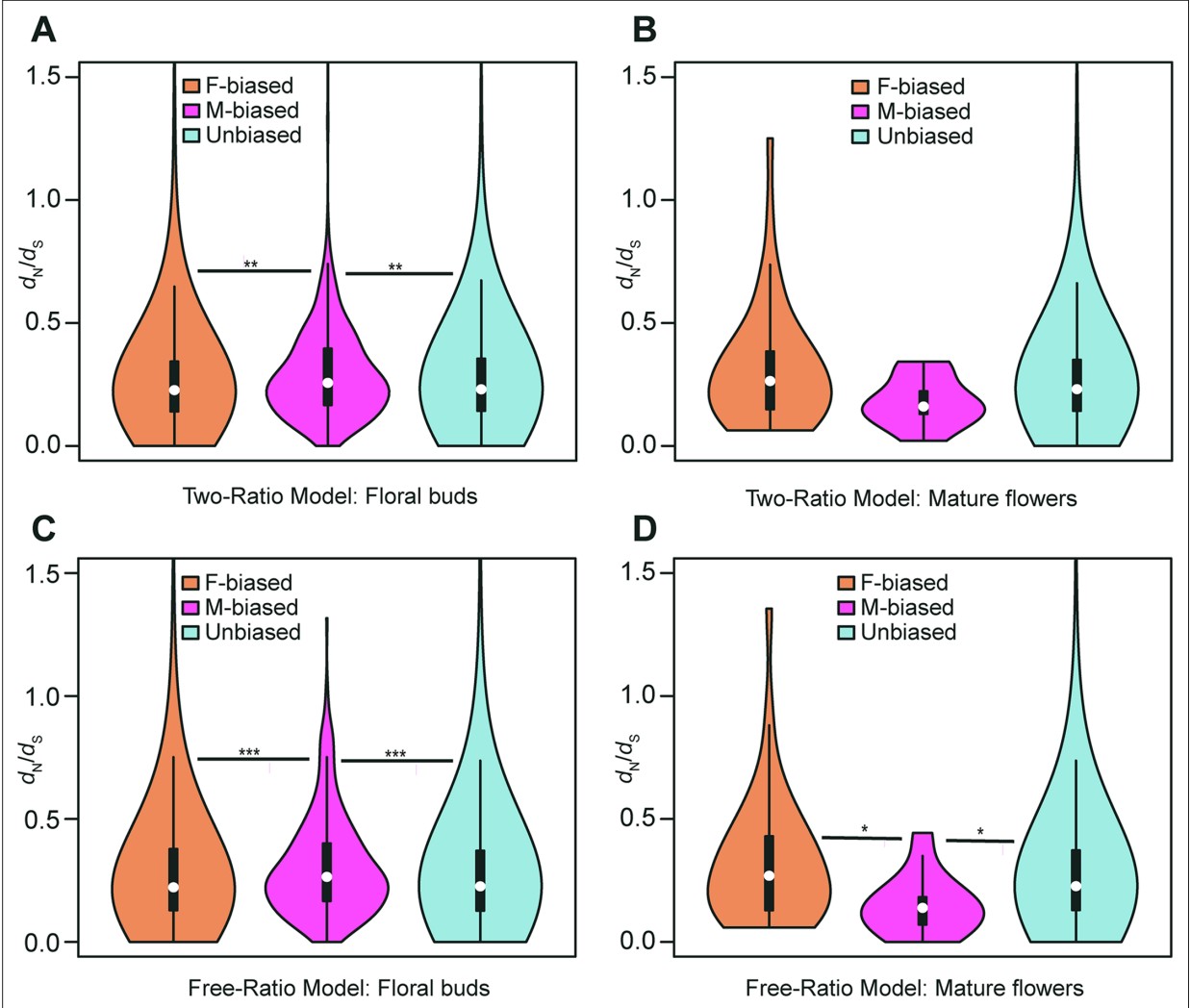

**Figure 4.** Violin plots of $d_N/d_S$ values ($0<\omega<2$) of female-biased, male-biased, and unbiased genes in floral buds and flowers at anthesis of *Trichosanthes pilosa*. White dot indicates the median of $d_N/d_S$ values for sex-biased and unbiased genes. Wilcoxon rank sum tests are used to test for significant differences (\*\*\*p<0.0005, \*\*p<0.005, and \*p<0.05). The distributions of $d_N/d_S$ values for female-biased, male-biased and unbiased genes in floral buds (**A**) and flowers at anthesis (**B**) using 'two-ratio' branch model. The distributions of $d_N/d_S$ values for female-biased, male-biased and unbiased genes in floral buds (**C**) and flowers at anthesis (**D**) using 'free-ratio' branch model.

The online version of this article includes the following figure supplement(s) for figure 4:

**Figure supplement 1.** Boxplot of $d_N/d_S$ values (including all $\omega$ values) of female-biased, male-biased, and unbiased genes in floral buds and flowers at anthesis of *Trichosanthes pilosa*.

**Figure supplement 2.** Violin plots of effective number of codons (ENCs) values of female-biased, male-biased and unbiased genes in floral buds.

**Figure supplement 3.** Boxplot of $d_S$ values of female-biased, male-biased, and unbiased genes in floral buds of dioecious *Trichosanthes pilosa* using 'free-ratio' (**A**) and 'two-ratio' (**B**) branch model.

found that only in floral buds, there were significant differences in $\omega$ values in the results of 'free-ratio' model (female-biased versus male-biased genes, p=0.04282 and male-biased versus unbiased genes, p=0.01114) and 'two-ratio' model (female-biased versus male-biased genes, p=0.01992 and male-biased versus unbiased genes, p=0.02127, respectively) by permutation t-test, which is consistent with the results of Wilcoxon rank sum test.

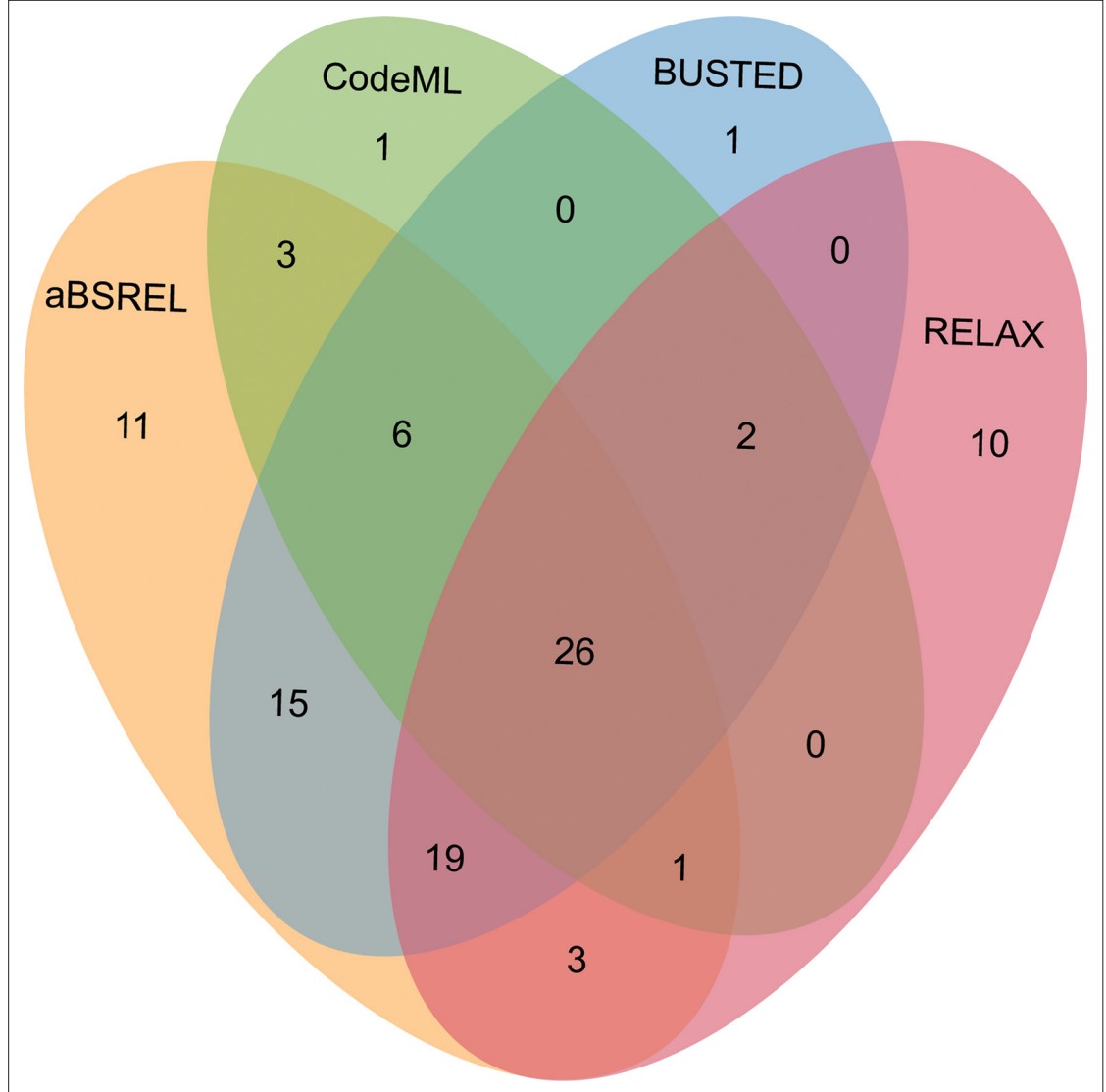

**Figure 5.** Venn diagrams of male-biased genes detected to be under positive selection using aBSREL, BUSTED, CodeML, and RELAX in floral buds.

The online version of this article includes the following figure supplement(s) for figure 5:

**Figure supplement 1.** Venn diagrams of female-biased genes under positive selection in floral buds.

**Figure supplement 2.** Venn diagrams of unbiased genes under positive selection in floral buds.

**Figure supplement 3.** Scatterplots of KEGG pathway of sex-biased genes in female (**A**) and male (**B**) floral buds of the dioecious *Trichosanthes pilosa*.

## Evidence of positive selection and relaxed selection for male-biased genes in floral buds

After comparing the alternative hypothesis (branch-site model A with estimated $\omega$ value) against the null model (branch-site model A with fixed $\omega=1$) (see Methods section), we discovered that 39 out of 343 OGs (11.34%) in male-biased genes of floral buds exhibited strong evidence of having certain sites that evolved under positive selection based on foreground $\omega$ value, likelihood ratio tests (LRTs, p<0.05) and Bayes empirical Bayes (BEB) value (*Figure 5* and *Supplementary file 6*). As a complementary approach, we utilized the aBSREL and BUSTED methods that are implemented in HyPhy v.2.5 software, which avoids false positive results by classical branch-site models due to the presence of rate variation in background branches, and detected significant evidence of positive selection. According to our findings, 84 out of 343 OGs (24.49%) were identified to be under episodic positive selection in male-biased genes of floral buds with a site proportion of 0.17–26.44% based on aBSREL

(*Supplementary file 7*). In addition, 69 out of 343 OGs (20.01%) exhibited significant signs of positive selection with the site proportion of 0.28–32.65% in male-biased genes of floral buds according to BUSTED (*Supplementary file 8*). Among these, a total of 32 OGs (9.30%) were identified through our tests using CodeML, aBSREL, and BUSTED (*Figure 5*).

Relaxed selection may occur when the efficiency of natural selection (e.g. the reduction of the strength of purifying selection) is reduced, leading to accumulations of deleterious mutations (*Lahti et al., 2009*; *Dapper and Wade, 2020*). This has been proposed as an explanation for the rapid evolution of sex-biased genes (*Lahti et al., 2009*; *Mank, 2017*). Using the RELAX model, we detected that 18 out of 343 OGs (5.23%) showed significant evidence of relaxed selection (K=0.0184–0.6497) (*Supplementary file 9*). Most of the 18 OGs were members of different gene families generated by gene duplication (*Supplementary file 13*). Additionally, we observed that 61 out of 343 OGs (17.73%) exhibited significant evidence of intensified positive selection (K=2.3363–50, $\omega_2 \geq 1$) (*Figure 5* and *Supplementary file 10*), which is consistent with the results obtained from CodeML, aBSREL, and BUSTED.

According to previous studies (*Ellegren and Parsch, 2007*; *Catalán et al., 2018*), genes that exhibit sex-biased expression with rapid evolutionary rates tend to display a lower codon bias compared to unbiased genes. In our results, we found that male-biased genes in floral buds had a significantly lower median effective number of codons (ENCs) than both female-biased and unbiased genes (Wilcoxon rank sum test, female-biased vs male-biased genes, p=0.0001 and male-biased vs unbiased genes, p=0.0123). This suggested that male-biased genes in floral buds exhibit stronger codon bias than both female-biased and unbiased genes (*Figure 4—figure supplement 2*). Similarly, given that $d_N/d_S$ values of sex-biased genes were higher due to codon usage bias, lower $d_S$ rates would be expected in sex-biased genes relative to unbiased genes (*Ellegren and Parsch, 2007*; *Parvathy et al., 2022*). However, we exhibited that the median of $d_S$ values in male-biased genes was much higher than those in female-biased and unbiased genes in the results of 'free-ratio' (*Figure 4—figure supplement 3*, female-biased versus male-biased genes, p=6.444e-12 and male-biased versus unbiased genes, p=4.564e-13) and 'two-ratio' model (*Figure 4—figure supplement 3*, female-biased versus male-biased genes, p=2.2e-16 and male-biased versus unbiased genes, p=9.421e-08, respectively). In short, our analyses indicated that rapid evolutionary rates of male-biased genes in floral buds were not associated with a reduction in codon usage bias.

We also analyzed whether female-biased and unbiased genes underwent positive and relaxed selection in floral buds (*Supplementary file 6–10*). We identified 216 (18.86%) positively selected (*Figure 5—figure supplement 1*), and 69 (6.03%) relaxed selective female-biased genes from 1145 OGs, respectively. Similarly, we found 436 (18.33%) positively selected (*Figure 5—figure supplement 2*), and 43 (1.81%) unbiased genes under relaxed selection from 2378 OGs, respectively. Notably, male-biased genes have a higher proportion (10%) of positively selected genes compared to female-biased and unbiased genes. However, relaxed selective male-biased genes have a higher proportion (3.24%) than unbiased genes, but about 0.8% lower than that of female-biased genes. In summary, our analyses suggested that positive selection and relaxed selection likely drove the rapid evolutionary rates of male-biased genes compared to female-biased and unbiased genes in floral buds.

## Functional analysis of sex-biased genes in floral buds

We conducted KEGG pathway enrichment analysis on sex-biased genes in floral buds. Our results showed that 699 genes were female-biased and 358 genes were male-biased, with significant enrichment (p<0.05) in 26 and 24 KEGG pathways, respectively (*Supplementary file 11*). In the floral bud stage, we observed that female-biased genes were mainly enriched in metabolic and signaling pathways, such as ribosome, Fatty acid elongation, photosynthesis, and plant hormone signal transduction (*Figure 5—figure supplement 3* and *Supplementary file 11*). On the other hand, male-biased genes were significantly enriched in metabolic and signaling pathways, including inositol phosphate metabolism, starch, and sucrose metabolism, regulation of autophagy, plant hormone signal transduction, and Toll-like receptor signaling pathway (*Figure 5—figure supplement 3* and *Supplementary file 11*).

We have also found that certain male-biased genes, which are evolving under positive selection and relaxed selection (*Supplementary file 12 and 13*), were related to abiotic stress and immune responses. For instance, mitogen-activated protein kinase kinase kinase 18 (MAPKKK18) (*Zhang and*

*Zhang, 2022*), zinc finger CCCH domain-containing protein 20 (C3H20/TZF2) (*Bogamuwa and Jang, 2014*), and heat stress transcription factor B-3 (HSFB3) (*Scharf et al., 2012*) have been linked to stress. Additionally, ten male-biased genes with rapid evolutionary rates were associated with anther and pollen development. These genes include LRR receptor-like serine/threonine protein kinase (LRR-RLK) (*Cui et al., 2022*), pollen receptor-like kinase 3 (*PRK3*) (*Muschietti and Wengier, 2018*), autophagy-related protein 18 f (*ATG18f*) (*Zhou et al., 2015*; *Li et al., 2020*), and plant homeodomain (PHD) finger protein 3 (*MALE STERILITY 3*) (*Hou et al., 2022*) in floral buds of male plants.

## Discussion

The Cucurbitaceae family, where half of the species are monoecious and half are dioecious, is an excellent model for studying the evolution of sexual systems of angiosperms, including sex-determination mechanism and sexual dimorphism (*Schaefer and Renner, 2010*; *Boualem et al., 2015*; *Ma and Pannell, 2016*). In this study, we compared the expression profiles of sex-biased genes between sexes and two tissue types, investigated whether sex-biased genes exhibited evidence of rapid evolutionary rates of protein sequences and identified the potential evolutionary forces responsible for the observed patterns in the dioecious *Trichosanthes pilosa*.

### Sex-biased expression in floral buds

Several studies have shown that in dioecious plants, male-biased genes tend to outnumber female-biased genes, consistent with the patterns in most animals (*Zhang et al., 2007*; *Djordjevic et al., 2022*). For instance, insect-pollinated dioecious plants such as *Asparagus officinalis* (*Harkess et al., 2015*) and *Silene latifolia* (*Zemp et al., 2016*), exhibit a higher proportion of male-biased genes. In contrast, the wind-pollinated dioecious plant *Populus balsamifera* (*Sanderson et al., 2019*) has twice as many female-biased genes as male-biased genes. The differences in these studies could be partly attributed to the impact of sexual selection on secondary sexual traits in insect-pollinated dioecious plants, as opposed to wind-pollinated ones (*Delph and Herlihy, 2012*; *Muyle, 2019*; *Sanderson et al., 2019*). Similar to the above study of *Populus balsamifera*, our findings revealed that the number of female-biased genes in floral buds of the night-flowering, insect-pollinated dioecious plant *Trichosanthes pilosa* exceeded that of male-biased genes by 882 (~21%). This excess of female-biased expression could be due to lower energy consumption needs and reduced chemical defense capability against insect herbivores in short-lived male flowers (*Sanderson et al., 2019*). Indeed, functional enrichment analysis in chemical pathways such as terpenoid backbone and diterpenoid biosynthesis indicated that relative to male floral buds, female floral buds had more expressed genes that were equipped to defend against herbivorous insects and pathogens, except for growth and development (*Vaughan et al., 2013*; *Ren et al., 2022*; *Figure 5—figure supplement 3* and *Supplementary file 11* ). Additionally, our enrichment analysis showed that the photosynthesis, porphyrin, and chlorophyll metabolism pathways were more active in female floral buds compared to male floral buds (*Figure 5—figure supplement 3* and *Supplementary file 11*), enabling them to acquire more resources such as carbon for fruit and seed production (*Delph, 1999*).

We identified functional enrichments in Toll-like receptor signaling, NF-kappa B signaling, and inositol phosphate metabolism pathways in male floral buds (*Figure 5—figure supplement 3* and *Supplementary file 11* ). We also found that male-biased genes with high evolutionary rates in male floral buds were associated with functions to abiotic stresses and immune responses (*Supplementary file 12 and 13*), which suggests that male floral buds through rapidly evolving genes are adapted to mountain climate and the environment in Southwest China relative to female floral buds through high gene expression. In addition, the enrichment in regulation of autophagy pathways could be associated with gamete development and the senescence of male floral buds (*Supplementary file 14*; *Liu and Bassham, 2012*; *Li et al., 2020*; *Zhou et al., 2021*). In fact, it was observed that male flowers senesced faster (*Wu et al., 2011*). We also found that homologous genes of two male-biased genes in floral buds (*Supplementary file 14*) that control the raceme inflorescence development (*Teo et al., 2014*) were highly expressed compared to female floral buds. Taken together, these results indicate that expression changes in sex-biased genes, rather than sex-specific genes play different roles in sexual dimorphic traits in physiology and morphology (*Dawson and Geber, 1999*).

## Rapid evolution of male-biased genes in floral buds

It has been observed that, in most animals, sex-biased genes, particularly those biased towards males, often exhibit more rapid evolutionary rates than unbiased genes (*Parsch and Ellegren, 2013*; *Grath and Parsch, 2016*; *Mank, 2017*; *Toubiana et al., 2021*). However, in dioecious angiosperms, no evidence of rapid evolution in sex-biased genes relative to unbiased genes has been found (*Zemp et al., 2016*; *Darolti et al., 2018*; *Cossard et al., 2019*; *Sanderson et al., 2019*; *Scharmann et al., 2021*). In contrast, our findings indicated that male-biased genes experience higher evolutionary rates than both female-biased and unbiased genes in floral buds of dioecious *T. pilosa*. We proposed that positive selection and relaxed purifying selection may be responsible for the rapid sequence evolution of male-biased genes.

After analyzing the data, we found that around 28.57% (98 genes) of male-biased genes have undergone positive selection. Additionally, we observed that the proportion of male-biased genes under positive selection was about 10% higher than that of female-biased and unbiased genes. Furthermore, we discovered that some male-biased genes under positive selection were linked to abiotic stress and immune responses (*Supplementary file 12*). Our findings are consistent with studies on *Drosophila* and *Ectocarpus* (*Zhang and Parsch, 2005*; *Lipinska et al., 2015*), suggesting that adaptive evolution is one of the important driving forces for rapid evolutionary rates. Notably, we identified several male-biased genes under positive selection that are functionally related to early flowering (*phyB*) (*Stephenson and Bertin, 1983*; *Forrest, 2014*; *Hajdu et al., 2015*) and pollen development (*Skogsmyr and Lankinen, 2002*; *Williams and Reese, 2019*; *Supplementary file 12–14*). These findings indicate that a small fraction of male-biased genes may experience adaptive evolution due to sexual selection, driven by male-male competition.

Alternatively, relaxed constraints could contribute to the rapid evolutionary rates of sex-biased genes through three key characteristics (*Dapper and Wade, 2020*; *Tosto et al., 2023*). First, sex-biased genes are often expressed solely in reproductive tissues of one sex (e.g. sex-specific genes), particularly in the haploid phase (*Sandler et al., 2018*; *Immler, 2019*; *Beaudry et al., 2020*). Sex-specific selection (e.g. relaxed purifying selection) acting on sex-specific genes could decrease the elimination of deleterious mutations (*Mank, 2017*), such as pollen-specific (*Harrison et al., 2019*; *Arunkumar et al., 2013*) or testes-specific genes (*Gershoni and Pietrokovski, 2014*). However, we observed male-biased genes but not male-specific genes undergoing relaxed purifying selection. Second, sex-biased genes are often expressed in few tissues (tissue-biased genes) (*Meisel, 2011*; *Tosto et al., 2023*), resulting in these genes rapidly evolving under positive selection or relaxed purifying selection due to low evolutionary constraints (*Congrains et al., 2018*; *Whittle et al., 2021*; *Tosto et al., 2023*). In our results, 343 male-biased genes (M1-biased genes, M1BGs) with faster evolutionary rates relative to female-biased and unbiased genes overlapped with 1755 tissue-biased genes in floral buds (M1-tissue-biased genes, M1TGs) (27 out of 343, 7.87%) (*Figure 3—figure supplement 1*). Furthermore, 27 out of 343 male-biased genes (that is, tissue-biased genes) in floral buds overlapped with nine out of 98 (9.18%) male-biased genes under positive selection (*Figure 3—figure supplement 1*), and one out of 18 (5.56%) male-biased genes under relaxed purifying selection (*Figure 3—figure supplement 1*). So, we obtained ten rapidly evolving tissue-biased genes which were also male-biased in male flower buds, suggesting that elevated evolutionary rates may partly be linked to low constraints, consistent with male-biased genes in *Anastrepha* and *Fucus* (*Congrains et al., 2018*; *Hatchett et al., 2023*). Finally, gene duplication has long been thought to promote functional divergences and phenotypic novelties by relaxing the constraints of purifying selection on the duplicated gene copy early in its history (*Lynch and Conery, 2000*; *Lynch and Katju, 2004*; *Lahti et al., 2009*). For instance, the progesterone receptor gene family in the human lineage *Marinić and Lynch, 2020* and the CYP98A9 clade in Brassicales (*Liu et al., 2016*) have demonstrated rapid evolution and divergent function due to relaxed purifying selection. In our results, we identified only 18 out of 343 (5.25%) male-biased genes that underwent relaxed purifying selection using RELAX model (*Supplementary file 13*). Interestingly, the vast majority of genes under relaxed selection were members of different gene families generated by gene duplication (including whole-genome duplication), such as LOB domain-containing protein 18 (*LBD18*) (*Zhang et al., 2020*), WRKY transcription factor 72 (*WRKY72*) (*Chen et al., 2017*), and pollen receptor-like kinase 3 (*PRK3*) (*Muschietti and Wengier, 2018*).

Reducing codon usage bias could theoretically accelerate evolutionary rates of sex-biased genes by decreasing synonymous substitution rates. However, our results did not support this idea due to stronger codon usage bias in male-biased genes (*Figure 4—figure supplement 2*). Codon usage bias is influenced by many factors, such as levels of gene expression. Highly expressed genes have a stronger codon usage bias and could be encoded by optimal codons for more efficient translation (*Frumkin et al., 2018*; *Parvathy et al., 2022*), consistent with high levels of gene expression in males (that is, male-biased genes) in floral buds. Additionally, stronger codon usage bias may be related to higher synonymous substitution rates (*Parvathy et al., 2022*). Indeed, male-biased genes had significantly higher median $d_S$ values than female-biased and unbiased genes, both in the 'free-ratio' analysis (*Figure 4—figure supplement 3*) and 'two-ratio' branch model (*Figure 4—figure supplement 3*).

The presence of sex chromosomes may be a potential confounding factor for evolutionary rates of sex-biased genes which are X-linked, Y-linked, and autosomal genes (*Hough et al., 2014*; *Sandler et al., 2018*). We distinguished these sex-biased genes on sex chromosomes from autosomal chromosomes following the steps of *Sandler et al., 2018*, and computed the overall comparable proportions of sex-linked genes among male-biased (3/343=0.087%), female-biased (19/1145=1.66%) and unbiased genes (36/2378=1.51%). These analyses suggested that sex-linked genes may contribute relatively little to the rapid evolution of male-biased genes.

Several species have been observed to exhibit rapid evolutionary rates of sequences on sex chromosomes compared to autosomes, which has been related to the evolutionary theories of fast-X or fast-Z (*Meisel and Connallon, 2013*; *Hough et al., 2014*; *Wright et al., 2015*; *Charlesworth et al., 2018*; *Darolti et al., 2023*). Furthermore, the quantification of gene expression by bulk RNA-seq technology, relative to single-cell transcriptome analysis, has been shown to potentially obfuscate true signals in the evolution of sex-biased gene expression in complex aggregations of diverse cell types (*Darolti and Mank, 2023*; *Tosto et al., 2023*). Additionally, our samples were relatively small, and may provide low power to detect differential expression and evolutionary analysis. Therefore, investigation of these interesting issues related to sex-biased gene evolution in *T. pilosa* can only be conducted when whole genome sequences and population datasets become available in the near future.

## Methods

### Plant materials and RNA isolation

Floral buds (≤3 mm) and flowers at anthesis were sampled from three female and three male plants (*Figure 1*) from the mountainous regions of Anning (Qinglong Gorge), Yunnan Province in Southwest China. Floral buds from female and male plants were named F1 and M1, respectively. Similarly, flowers at anthesis from female and male plants were named F2 and M2, respectively (*Supplementary file 1*). To exclude possible bacterial contamination, all tissues were sterilized with 75% alcohol and immediately rinsed with purified water. All samples were then snap-frozen in liquid nitrogen, and stored at –80 °C. Total RNA was extracted from each sample using TRIzol reagent (Life Technologies, CA, USA) according to the manufacturer's instructions. The quantification and qualification of RNA were assessed by the RNA Nano 6000 Assay Kit of the Bioanalyzer 2100 system (Agilent Technologies, CA, USA).

### Illumina sequencing, de novo assembly, and annotation

To construct the library, approximately 2 μg of total RNA was used with the Illumina NEBNext UltraTM RNA Library Prep Kit. RNA sequencing was performed on the Illumina NovaSeq 6000, generating 150 bp paired-end reads. The resulting clean reads were obtained by removing adapters, reads containing N bases and low-quality reads using Trimmomatic v.0.39 (*Bolger et al., 2014*). These reads were deposited in the NCBI database (PRJNA899312).

De novo assembly for clean reads from all samples was performed using Trinity v.2.10.0 (*Haas et al., 2013*) with min_kmer_cov: 3 and all other default parameters. To eliminate contamination, all transcripts of de novo assembly were compared to bacterial genomes downloaded from NCBI databases using BLASTN with an e-value of 1.0e-05 in blast + 2.12.0 software. We used Corset v.4.6 (*Davidson and Oshlack, 2014*) to obtain high-quality, non-redundant consensus transcripts (unigenes). TransDecoder v.5.5.0 was run with -m 100 parameters, namely at least 100 amino acids, to predict the coding DNA and protein sequences (*Haas et al., 2013*).

To evaluate the accuracy and completeness of reference transcriptomes, we performed gene function annotations based on the following databases, using BLAST with a cutoff e-value of 1.0e-05: NR, NT, and Swissport (*Shiryev et al., 2007*). We mapped the unigenes to Pfam database using InterProScan v.5.41 (*Jones et al., 2014*), to the GO database using Blast2GO (*Conesa et al., 2005*), and to the KEGG database using KEGG automatic annotation server (*Moriya et al., 2007*). Additionally, we estimated the completeness of reference transcriptomes using BUSCO v.5.4.5 based on embryophyta_odb10 database (*Seppey et al., 2019*).

## Detection of sex-biased genes

Clean reads were mapped onto all unigenes using Bowtie2 (*Langmead and Salzberg, 2012*). Read counts were normalized to FPKM (Fragments Per Kilobase Million) value for each unigene using RSEM (*Li and Dewey, 2011*) in different male and female samples. Genes with zero read counts (i.e. no expression) in both two sexes and tissues were excluded. Differential expression analysis between sexes and tissue types was performed using DESeq2 R package (*Love et al., 2014*). Unigenes with an FDR-adjusted $p<0.05$ and an absolute value of $\log_2$ ratio $\geq 1$ identified by DESeq2 were considered as sex-biased genes. To perform KEGG functional enrichment, we used all KEGG annotation terms for all genes as the background and performed the analyses using KOBAS v.2.0.12 (*Mao et al., 2005*).

## Evolutionary rate analyses

To quantify the evolutionary rates of sex-biased genes, we download published genome datasets for monoecious *Trichosanthes anguina* (*Ma et al., 2020*) and monoecious *Luffa cylindrica* which has a closer phylogenetic relationship with *Trichosanthes* (*de Boer et al., 2012*; *Wu et al., 2020*) from CuGenDB database (*Zheng et al., 2019*). Additionally, we also download published RNA sequencing reads of floral buds and flowers from CNCB (Accession CRA002313) and NCBI databases (Accession SRR5259239) for dioecious plant *Trichosanthes kirilowii* (*Hu et al., 2020*), and de novo assembled by previously described methods.

We identified one-to-one OGs using OrthoFinder v.2.3.3 with default parameters from *T. anguina*, *T. pilosa*, *T. kirilowii*, and *Luffa cylindrica* (*Emms and Kelly, 2019*). Then, we employed TranslatorX with -c 1 p M -g -b5 n parameters (i.e. the multiple alignment and the trimming using Muscle and GBlocks, respectively), translated nucleotide sequences and back-translated amino acid alignments into nucleotide alignments to ensure codon-to-codon alignment (*Abascal et al., 2010*). The remaining gapless alignments ($\geq 100$ bp in length) were retained.

To investigate the evolutionary rates of coding sequences, we estimated nonsynonymous substitution ($d_N$), synonymous substitution ($d_S$) rates, as well as protein substitution rates ($d_N/d_S$, $\omega$), using two branch models from CodeML package in PAML v.4.9h with the F3 × 4 codon frequencies (CodonFreq = 2) (*Yang, 2007*). According to the phylogenetic relationships of *Trichosanthes* (*de Boer et al., 2012*; *Guo et al., 2020*), we set up tree structures ((*T. anguina*, *T. pilosa*), *T. kirilowii*, *L. cylindrica*) in the control file of CodeML. First, we employed a 'two-ratio' branch model (model = 2, Nssites = 0) that assumes the foreground (two dioecious species) has a different $\omega$ value from the background (two monoecious species) to estimate and compare the divergences of the foreground. Second, to reduce the potential bias of $\omega$ value due to the conflation of two dioecious species, we also implemented a 'free-ratio' branch model (model = 1, Nssites = 0), which assumes an independent $\omega$ ratio for each branch. Finally, to avoid the effects of saturation substitution, we used separately OGs with $0<\omega<2$ and all OGs with $\omega>0$, plotted the distribution of $\omega$ values, and compared the median of $\omega$ values in female-biased, male-biased and unbiased orthologous genes of floral buds and flowers at anthesis. All comparisons between sex-biased and unbiased genes were tested using the Wilcoxon rank sum test in R software. Additionally, we also performed permutations t-tests with 100,000 permutations in the R package Deducer (*Fellows, 2012*).

## Estimation of the strength of natural selection

The rapid evolutionary rates of sex-biased genes may be attributed to positive selection, relaxed selection, and lower codon usage bias (*Catalán et al., 2018*; *Dapper and Wade, 2020*). Therefore, we conducted separate analyses using classical branch-site models that assume different $\omega$ values both among branches and across sites (*Álvarez-Carretero et al., 2023*), the adaptive branch-site random effects likelihood (aBSREL) model (*Smith et al., 2015*), the branch-site unrestricted statistical test for

episodic diversification (BUSTED) model (*Murrell et al., 2015*), the RELAX model (*Wertheim et al., 2015*), and the effective number of codons (ENC) in PAML v.4.9h (*Yang, 2007*), HyPhy v.2.5 (*Kosakovsky Pond et al., 2020*) and CodonW v.1.4.2 (http://codonw.sourceforge.net) to distinguish which evolutionary forces are driving the rapid evolutionary rates of sex-biased genes.

To determine if amino acid sites in the foreground, including the *T. pilosa* lineage have undergone positive selection (foreground $\omega > 1$) compared with the background for each OGs, we followed the steps of *Zhang et al., 2005*, and used branch-site model A (model = 2, Nssite = 2, fix_omega = 0, omega = 1.5) and branch-site model null (model = 2, Nssite = 2, fix_omega = 1, omega = 1). The classical branch-site model assumes four site classes (0, 1, 2 a, 2b), with different $\omega$ values for the foreground and background branches. In site classes 2 a and 2b, the foreground branch undergoes positive selection when there is $\omega > 1$. We examined the significance of likelihood ratio tests (LRTs, $p < 0.05$) to identify positively selected sites between model A and model null by comparing LRTs to the Chi-square distribution with two degrees of freedom. We adjusted the LRTs p value for multiple comparisons using Benjamini and Hochberg's (FDR) algorithm. When the p value was significant, we used Bayes Empirical Bayes (BEB) estimates to identify sites with a high posterior probability (pp ≥0.95) of being under positive selection (*Yang et al., 2005*).

To detect episodic positive selection at a proportion of sites on the foreground branch, we employed the aBSREL method (*Smith et al., 2015*) in the HyPhy v.2.5 packages to compare the fully adaptive model ($\omega > 1$) to the null model that allows no positive selection rate classes by LRTs, which is an improved algorithm of branch-site models in PAML. For relatively small datasets, such as those with fewer than 10 taxa, the aBSREL method may not have enough power to detect positive selection. Therefore, we also ran the BUSTED method to identify gene-wide evidence of episodic positive selection at least one site on at least one branch (*Murrell et al., 2015*). We set *T. pilosa* as the foreground and assessed the statistical significance ($p < 0.05$) using LRTs with the Holm-Bonferroni correction.

To test the relaxation of selective strength, we utilized the RELAX model in the HyPhy v.2.5 software (*Wertheim et al., 2015*; *Schrader et al., 2021*). The RELAX model estimates three $\omega$ parameters ($\omega_0 \leq \omega_1 \leq 1 \leq \omega_2$), and determines the proportion of sites in the test (foreground) and reference (background) branches using a branch-site model. The first two $\omega$ classifications indicate that sites have undergone purifying selection, and the third classification indicates that sites have been under positive selection. Additionally, the model introduces a selection intensity parameter (K value) to compare a null model (K=1) with an alternative model, thereby assessing the strength of natural selection. When K>1, it suggests intensified natural selection, when K<1, indicates relaxed natural selection in the test branch relative to the reference branch. We quantified the statistical confidence of K value ($p < 0.05$) using LRTs and the Holm-Bonferroni correction.

To investigate codon usage bias, which refers to the differences in the frequency of occurrence of synonymous codons in coding DNA, we employed CodonW v.1.4.2. This program considers the ENC values from 20 to 61 as a measure of the departure of the genetic codes for a given gene (*Wright, 1990*), with lower ENC values represent stronger codon usage bias (*Hambuch and Parsch, 2005*). We performed a Wilcoxon rank sum test to determine if there were deviations in ENC values among female-biased, male-biased, and unbiased genes in floral buds.

## Acknowledgements

We are very grateful to Spencer C H Barrett (University of Toronto, Canada) for his critical reading and suggestions on the manuscript. We also thank three anonymous reviewers for their comments on improving the quality of the manuscript. We are indebted to Ting Zhang, Zhi-Yun Yang, Jiang-Li Ma, Peng-Fei Ma, Xu-Kun Wu, Shi-Yu Lv, Zhen Peng, and other members of staff of Germplasm Bank of Wild Species for sampling. We also thank the iFlora HPC Center (iFlora High-Performance Computing Center) of Germplasm Bank of Wild Species for computational support on data analysis.

# Additional information

## Funding

| Funder | Grant reference number | Author |
|---|---|---|
| Chinese Academy of Sciences (CAS) | Strategic Priority Research Program (XDB 31000000) | De-Zhu Li |
| National Natural Science Foundation of China | 32370233 | Hong-Tao Li |
| Key R & D Program of Yunnan Province, China | 202103AC100003 | De-Zhu Li |
| Key Basic Research Program of Yunnan Province, China | 202101BC070003 | Hong-Tao Li |
| Science and Technology Basic Resources Investigation Program of China | 2019FY100900 | Hong-Tao Li |
| Open Research Project of "Cross-Cooperative Team" of the Germplasm Bank of Wild Species, CAS Kunming Institute of Botany | | Hong-Tao Li |
| National Wild Plant Germplasm Resource Center, China | | Hong-Tao Li |
| CAS Key Technology Talent Program | | Hong-Tao Li |
| National Natural Science Foundation of China | 31570333 | Hong-Tao Li |

The funders had no role in study design, data collection and interpretation, or the decision to submit the work for publication.

## Author contributions

Lei Zhao, Conceptualization, Resources, Data curation, Software, Formal analysis, Supervision, Investigation, Visualization, Methodology, Writing - original draft, Writing – review and editing; Wei Zhou, Writing – review and editing; Jun He, Investigation; De-Zhu Li, Conceptualization, Supervision, Funding acquisition, Project administration, Writing – review and editing; Hong-Tao Li, Conceptualization, Resources, Data curation, Supervision, Funding acquisition, Investigation, Project administration, Writing – review and editing

## Author ORCIDs

Lei Zhao ⓘ http://orcid.org/0000-0002-2253-5319
De-Zhu Li ⓘ https://orcid.org/0000-0002-4990-724X
Hong-Tao Li ⓘ https://orcid.org/0000-0002-1290-0917

Reviewer #1 (Public Review): https://doi.org/10.7554/eLife.89941.5.sa1
Reviewer #2 (Public Review): https://doi.org/10.7554/eLife.89941.5.sa2
Reviewer #3 (Public Review): https://doi.org/10.7554/eLife.89941.5.sa3
Author Response https://doi.org/10.7554/eLife.89941.5.sa4

# Additional files

## Supplementary files

• Supplementary file 1. Overview of sequencing reads from 12 samples of male and female plants in

*Trichosanthes pilosa.*

- Supplementary file 2. Numbers of unigenes annotated in public databases.

- Supplementary file 3. The mapping rate of reads for each sample in floral buds and flowers at anthesis of *Trichosanthes pilosa.*

- Supplementary file 4. $d_N$, $d_S$, and $\omega$ values of each female-biased, male-biased, unbiased orthologous genes of floral buds and flowers at anthesis for each species using 'two-ratio' branch model of CodeML in PAML. Two dioecious species *Trichosanthes pilosa* and *T. kirilowii* represent the foreground and two monoecious species *T. anguina* and *Luffa cylindrica* represent the background.

- Supplementary file 5. $d_N$, $d_S$, and $\omega$ values of each female-biased, male-biased, unbiased orthologous genes of floral buds and flowers at anthesis for each species using 'free-ratio' branch model of CodeML in PAML.

- Supplementary file 6. Genes under positive selection identified by branch-site model of CodeML in PAML and function in NR, KEGG, Swissport, and GO databases for male-biased orthologous genes in floral buds. Two dioecious species *Trichosanthes pilosa* and *T. kirilowii* represent the foreground and two monoecious species, *T. anguina* and *Luffa cylindrica* represent the background.

- Supplementary file 7. Genes under episodic positive selection tested by aBSREL model in HyPhy and functions in NR, KEGG, Swissport, and GO databases for male-biased orthologous genes in floral buds. The dioecious species *Trichosanthes pilosa* represents the foreground and other species represent the background.

- Supplementary file 8. Genes under episodic positive selection found by BUSTED model in HyPhy and functions in NR, KEGG, Swissport, and GO databases for male-biased orthologous genes in floral buds. The dioecious species *Trichosanthes pilosa* represents the foreground (unconstrained branch) and other species represent the background (constrained branch).

- Supplementary file 9. Genes under relaxed selection detected by RELAX model in HyPhy and functions in NR, KEGG, Swissport, and GO databases for male-biased orthologous genes in floral buds. The dioecious species *Trichosanthes pilosa* represents the foreground (test) and other species represent the background (reference).

- Supplementary file 10. Genes under intensified positive selection identified by RELAX model in HyPhy and functions in NR, KEGG, Swissport, and GO databases for male-biased orthologous genes in floral buds. The dioecious species *Trichosanthes pilosa* represents the foreground (test) and other species represent the background (reference).

- Supplementary file 11. KEGG pathway enrichment analysis of female-biased and male-biased genes in floral buds of the dioecious *Trichosanthes pilosa.*

- Supplementary file 12. Functions and references associated with abiotic stress and immune responses, organ developments of male-biased genes under significant positive selection (p<0.05) in floral buds.

- Supplementary file 13. Functions and references associated with abiotic stress and immune responses, organ developments of male-biased genes under significant relaxed selection (p<0.05) in floral buds.

- Supplementary file 14. The expressions and functions of some male-biased genes associated with senescence, raceme inflorescence development and early flowering in floral buds.

- MDAR checklist

- Source data 1. Reference transcriptome, orthology data, and alignments.

## Data availability

All RNA-Sequencing clean reads have been deposited in the databases of the National Center for Biotechnology Information (NCBI) under BioProject ID PRJNA899312. The reference transcriptome, orthology data, and alignments are available as *Source data 1*.

The following dataset was generated:

| Author(s) | Year | Dataset title | Dataset URL | Database and Identifier |
|-----------|------|---------------|-------------|-------------------------|
| Zhao L | 2022 | Trichosanthes pilosa Transcriptome or Gene expression | https://www.ncbi.nlm.nih.gov/bioproject/?term=PRJNA899312 | NCBI BioProject, PRJNA899312 |

The following previously published datasets were used:

| Author(s) | Year | Dataset title | Dataset URL | Database and Identifier |
|---|---|---|---|---|
| Xin J | 2020 | Transcriptome sequencing and screening of genes related to sex determination of Trichosanthes kirilowii Maxim | https://ngdc.cncb.ac.cn/gsa/browse/CRA002313 | Genome Sequence Archive, CRA002313 |
| University of Nebraska-Lincoln | 2017 | RNA-Seq of Trichosanthes kirilowii for transcriptome assembly | https://www.ncbi.nlm.nih.gov/sra/?term=SRR5259239 | NCBI Sequence Read Archive, SRR5259239 |

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
