## [Editor Report · eLife assessment]

This **valuable** paper examines gene expression differences between male and female individuals over the course of flower development in the dioecious angiosperm Trichosantes pilosa. Male-biased genes evolve faster than female-biased and unbiased genes, which is frequently observed in animals, but this is the first report of such a pattern in plants. In spite of the limited sample size, the evidence is mostly **solid** and the methods appropriate for a non-model organism. The resources produced will be used by researchers working in the Cucurbitaceae, and the results obtained advance our understanding of the mechanisms of plant sexual reproduction and its evolutionary implications: as such they will broadly appeal to evolutionary biologists and plant biologists.

---

## [Referee Report · Reviewer #1 (Public Review)]

The evolution of dioecy in angiosperms has significant implications for plant reproductive efficiency, adaptation, evolutionary potential, and resilience to environmental changes. Dioecy allows for the specialization and division of labor between male and female plants, where each sex can focus on specific aspects of reproduction and allocate resources accordingly. This division of labor creates an opportunity for sexual selection to act and can drive the evolution of sexual dimorphism.

In the present study, the authors investigate sex-biased gene expression patterns in juvenile and mature dioecious flowers to gain insights into the molecular basis of sexual dimorphism. They find that a large proportion of the plant transcriptome is differentially regulated between males and females with the number of sex-biased genes in floral buds being approximately 15 times higher than in mature flowers. The functional analysis of sex-biased genes reveals that chemical defense pathways against herbivores are up-regulated in the female buds along with genes involved in the acquisition of resources such as carbon for fruit and seed production, whereas male buds are enriched in genes related to signaling, inflorescence development and senescence of male flowers. Furthermore, the authors implement sophisticated maximum likelihood methods to understand the forces driving the evolution of sex-biased genes. They highlight the influence of positive and relaxed purifying selection on the evolution of male-biased genes, which show significantly higher rates of non-synonymous to synonymous substitutions than female or unbiased genes. This is the first report (to my knowledge) highlighting the occurrence of this pattern in plants. Overall, this study provides important insights into the genetic basis of sexual dimorphism and the evolution of reproductive genes in Cucurbitaceae.

---

## [Referee Report · Reviewer #2 (Public Review)]

Summary:

This study uses transcriptome sequence from a dioecious plant to compare evolutionary rates between genes with male- and female-biased expression and distinguish between relaxed selection and positive selection as causes for more rapid evolution. These questions have been explored in animals and algae, but few studies have investigated this in dioecious angiosperms, and none have so far identified faster rates of evolution in male-biased genes (though see Hough et al. 2014 https://doi.org/10.1073/pnas.1319227111).

Strengths:

The methods are appropriate to the questions asked. Both the sample size and the depth of sequencing are sufficient, and the methods used to estimate evolutionary rates and the strength of selection are appropriate. The data presented are consistent with faster evolution of genes with male-biased expression, due to both positive and relaxed selection.

This is a useful contribution to understanding the effect of sex-biased expression in genetic evolution in plants. It demonstrates the range of variation in evolutionary rates and selective mechanisms, and provides further context to connect these patterns to potential explanatory factors in plant diversity such as the age of sex chromosomes and the developmental trajectories of male and female flowers.

Weaknesses:

The presence of sex chromosomes is a potential confounding factor, since there are different evolutionary expectations for X-linked, Y-linked, and autosomal genes. Attempting to distinguish transcripts on the sex chromosomes from autosomal transcripts could provide additional insight into the relative contributions of positive and relaxed selection.

---

## [Referee Report · Reviewer #3 (Public Review)]

The potential for sexual selection and the extent of sexual dimorphism in gene expression have been studied in great detail in animals, but hardly examined in plants so far. In this context, the study by Zhao, Zhou et al. al represents a welcome addition to the literature.

Relative to the previous studies in Angiosperms, the dataset is interesting in that it focuses on reproductive rather than somatic tissues (which makes sense to investigate sexual selection), and includes more than a single developmental stage (buds + mature flowers).

---

## [Author Response]

The following is the authors’ response to the previous reviews.

**eLife assessment**
This valuable paper examines gene expression differences between male and female individuals over the course of flower development in the dioecious angiosperm Trichosantes pilosa. Male-biased genes evolve faster than female-biased and unbiased genes, which is frequently observed in animals, but this is the first report of such a pattern in plants. In spite of the limited sample size, the evidence is mostly solid and the methods appropriate for a non-model organism. The resources produced will be used by researchers working in the Cucurbitaceae, and the results obtained advance our understanding of the mechanisms of plant sexual reproduction and its evolutionary implications: as such they will broadly appeal to evolutionary biologists and plant biologists.
**Public Reviews:**

**Reviewer #1 (Public Review):**
The evolution of dioecy in angiosperms has significant implications for plant reproductive efficiency, adaptation, evolutionary potential, and resilience to environmental changes. Dioecy allows for the specialization and division of labor between male and female plants, where each sex can focus on specific aspects of reproduction and allocate resources accordingly. This division of labor creates an opportunity for sexual selection to act and can drive the evolution of sexual dimorphism.In the present study, the authors investigate sex-biased gene expression patterns in juvenile and mature dioecious flowers to gain insights into the molecular basis of sexual dimorphism. They find that a large proportion of the plant transcriptome is differentially regulated between males and females with the number of sex-biased genes in floral buds being approximately 15 times higher than in mature flowers. The functional analysis of sex-biased genes reveals that chemical defense pathways against herbivores are up-regulated in the female buds along with genes involved in the acquisition of resources such as carbon for fruit and seed production, whereas male buds are enriched in genes related to signaling, inflorescence development and senescence of male flowers. Furthermore, the authors implement sophisticated maximum likelihood methods to understand the forces driving the evolution of sex-biased genes. They highlight the influence of positive and relaxed purifying selection on the evolution of male-biased genes, which show significantly higher rates of non-synonymous to synonymous substitutions than female or unbiased genes. This is the first report (to my knowledge) highlighting the occurrence of this pattern in plants. Overall, this study provides important insights into the genetic basis of sexual dimorphism and the evolution of reproductive genes in Cucurbitaceae.
**Reviewer #2 (Public Review):**
Summary:This study uses transcriptome sequence from a dioecious plant to compare evolutionary rates between genes with male- and female-biased expression and distinguish between relaxed selection and positive selection as causes for more rapid evolution. These questions have been explored in animals and algae, but few studies have investigated this in dioecious angiosperms, and none have so far identified faster rates of evolution in male-biased genes (though see Hough et al. 2014 https://doi.org/10.1073/pnas.1319227111).Strengths:The methods are appropriate to the questions asked. Both the sample size and the depth of sequencing are sufficient, and the methods used to estimate evolutionary rates and the strength of selection are appropriate. The data presented are consistent with faster evolution of genes with male-biased expression, due to both positive and relaxed selection.This is a useful contribution to understanding the effect of sex-biased expression in genetic evolution in plants. It demonstrates the range of variation in evolutionary rates and selective mechanisms, and provides further context to connect these patterns to potential explanatory factors in plant diversity such as the age of sex chromosomes and the developmental trajectories of male and female flowers.Weaknesses:The presence of sex chromosomes is a potential confounding factor, since there are different evolutionary expectations for X-linked, Y-linked, and autosomal genes. Attempting to distinguish transcripts on the sex chromosomes from autosomal transcripts could provide additional insight into the relative contributions of positive and relaxed selection.
**Reviewer #3 (Public Review):**
The potential for sexual selection and the extent of sexual dimorphism in gene expression have been studied in great detail in animals, but hardly examined in plants so far. In this context, the study by Zhao, Zhou et al. al represents a welcome addition to the literature.Relative to the previous studies in Angiosperms, the dataset is interesting in that it focuses on reproductive rather than somatic tissues (which makes sense to investigate sexual selection), and includes more than a single developmental stage (buds + mature flowers).
**Recommendations for the authors:**

**Reviewer #3 (Recommendations For The Authors):**
I have reviewed this new version and find that it now addresses some of the shortcomings of the previous manuscript. However, several important limitations still remain:1. The conclusion that sex-linked genes contribute relatively little to the patterns described is important and would be worth including in the manuscript briefly (not just the response letter), focusing for instance on the overall comparable proportions of sex-linked genes among male-biased (3/343=0.087%), female-biased (19/1145=1.66%) and unbiased genes (36/2378=1.51%).

Authors’ response: Thank you for your advice. We have added these sentences in “Discussion” section (Lines 492-499).

2. The new sentence included in the results "we also found that most of them were members of different gene families generated by gene duplication" is too vague. The motivation of this analysis is not explained, leaving the intended message unclear.

Authors’ response: In the previous revision, as stressed by reviewer #1 “(2) Paragraph (407-416) describes the analysis of duplicated genes under relaxed selection but there is no mention of this in the results”, we added the sentence “we also found that most of them were members of different gene families generated by gene duplication” in “Relaxed selection” paragraph of the results. Accordingly, in “Discussion” section, we discussed the associations between gene duplication and relaxed selection (Lines 461-473).

Following your suggestion, we revised the results (Lines 304-307) to “Using the RELAX model, we detected that 18 out of 343 OGs (5.23%) showed significant evidence of relaxed selection (K = 0.0184–0.6497) (Tables S9). Most of the 18 OGs are members of different gene families generated by gene duplication (Table S13)”. This makes it more coherent with the discussion.

3. The sentences "given that dN/dS values of sex-biased genes were higher due to codon usage bias..." are very confusing. I do not understand the argument being made here. I do not see why "lower dS rates would be expected in sex-biased genes ..."

Authors’ response: We respectfully argue that codon usage bias was positively related to synonymous substitution rates. That is, stronger codon usage bias may be related to higher synonymous substitution rates (Parvathy et al., 2022). Lower ENC values represent stronger codon usage bias. So, if ω (dN/dS) values of sex-biased genes are higher due to codon usage bias, we expect lower dS rates (That is, higher ENC values). Please refer to the relevant papers (e. g. Darolti et al., 2018; Catalan et al., 2018; Schrader et al., 2021, cited in the references of the paper).

4. The manuscript now reports the proportion of unitigs annotated by similarity with a number of species. While this is an interesting observation, the reviewer was actually asking for a comparison between the number of unitigs (59,051) and the number of genes annotated in a typical cucurbitaceae genome. This would give an indication of the level of redundancy of the de novo assembled transcriptome.

Authors’ response: We admit that in the final assembly, transcripts may be overestimated. We respectfully suggest that it may be inappropriate to assess the redundancy of the de novo assembled transcriptome by comparing the transcriptome sequences with the genomic sequences. An appropriate approach is to compare transcriptome sequences and transcriptome sequences among different species. For example, Hu et al., 2020 (reference cited in the paper) obtained 145,975 non-redundant unigenes from flower buds of female and male plants in Trichosanthes kirilowii. Mohanty et al. (2017) obtained 71,823 non-redundant unigenes from flower buds of female and male plants in Coccinia grandis.

Reference:

Mohanty JN, Nayak S, Jha S, Joshi RK. 2017. Transcriptome profiling of the floral buds and discovery of genes related to sex-differentiation in the dioecious cucurbit Coccinia grandis (L.) Voigt. Gene. 626: 395-406.

5. From reading the text I could not understand the extent to which the permutation test actually agreed with the Wilcoxon rank sum test. The text says that the results were "almost consistent", which is too vague. This paragraph should be clarified.

Authors’ response: We performed permutation test for sex-biased genes in floral buds and flowers at anthesis. However, only in floral buds, the results of both tests (permutation test and Wilcoxon rank sum test) are significant. Taking your suggestions in consideration, we have revised them as “Additionally, we found that only in floral buds, there were significant differences in ω values in the results of ‘free-ratio’ model (female-biased versus male-biased genes, P = 0.04282 and male-biased versus unbiased genes, P = 0.01114) and ‘two-ratio’ model (female-biased versus male-biased genes, P = 0.01992 and male-biased versus unbiased genes, P = 0.02127, respectively) by permutation t test, which is consistent with the results of Wilcoxon rank sum test.(Lines 273-280)”.

6. The paragraph on the link between codon usage and dN/dS is very unclear and quite unnecessary. I would suggest to simply remove lines 312-323.

Authors’ response: We respectfully argue that codon usage bias is one of the most important factors for higher rates of sequence evolution. Please refer to Darolti et al. (2018), Catalan et al. (2018) and Schrader et al. (2021) (cited in the references of the paper). We retain these lines here.

7. The discussion contains many unnecessary repeats from the introduction and results section. I suggest shortening drastically at several places, including:remove lines 367-369

Authors’ response: Thank you for your suggestion. We revised these lines to “In this study, we compared the expression profiles of sex-biased genes between sexes and two tissue types, investigated whether sex-biased genes exhibited evidence of rapid evolutionary rates of protein sequences and identified the evolutionary forces responsible for the observed patterns in the dioecious Trichosanthes pilosa (Lines 369-373)”.

We removed the sentence “We compared the expression profiles of sex-biased genes between sexes and two tissue types and examined the signatures of rapid sequence evolution for sex-biased genes, as well as the contributions of potential evolutionary forces. (Lines 374-376)”.

remove lines 395-410

Authors’ response: Here we mainly discussed the possible associations between sex-biased genes, adaptation and sexual dimorphic traits. We retain them here for clarity.

remove lines 449-483, as they are almost entirely repetitions of elements already made clear in the results section.

Authors’ response: In these paragraphs, we discussed reasons that lead to relaxed purifying selection for sex-biased genes. They are coherent with the results section. We retain them to make it clearer.

Minor comments:line 146: remove "However"

Authors’ response: We have revised it.

line 187: "female flower buds tend to masculinize": the meaning is obscure

Authors’ response: We revised them as “Using hierarchical clustering analysis, we evaluated different levels of gene expression across sexes and tissues (Fig. 2C). Gene expression for female floral buds clustered most distantly from expression in female flowers at anthesis. However, expression in male floral buds clustered with expression in female flowers at anthesis, suggesting that male floral buds maybe tend to feminization in the early stages of floral development.”.

line 226: "we sequenced transcriptomes of T. pilosa": rather say "we used the transcriptomes described above for T. pilosa"

Authors’ response: We have revised it.

line 279: the meaning of "branch-site model A and branch site model null" is still not made clear.

Authors’ response: We have revised it.

line 324: change to: "we also analysed whether female-biased and unbiased genes underwent... "

Authors’ response: We have revised it.